# Two-pore channel blockade by phosphoinositide kinase inhibitors YM201636 and PI-103 determined by a histidine residue near pore-entrance

Canwei Du[1], Xin Guan[1] & Jiusheng Yan [1,2]✉

Human two-pore channels (TPCs) are endolysosomal cation channels and play an important role in NAADP-evoked $Ca^{2+}$ release and endomembrane dynamics. We found that YM201636, a PIKfyve inhibitor, potently inhibits $PI(3,5)P_2$-activated human TPC2 with an $IC_{50}$ of 0.16 μM. YM201636 also effectively inhibits NAADP-activated TPC2 and a constitutively-open TPC2 L690A/L694A mutant channel; whereas it exerts little effect when applied in the channel's closed state. PI-103, a YM201636 analog and an inhibitor of PI3K and mTOR, also inhibits human TPC2 with an $IC_{50}$ of 0.64 μM. With mutational, virtual docking, and molecular dynamic simulation analyses, we found that YM201636 and PI-103 directly block the TPC2's open-state channel pore at the bundle-cross pore-gate region where a nearby H699 residue is a key determinant for channel's sensitivity to the inhibitors. H699 likely interacts with the blockers around the pore entrance and facilitates their access to the pore. Substitution of a Phe for H699 largely accounts for the TPC1 channel's insensitivity to YM201636. These findings identify two potent TPC2 channel blockers, reveal a channel pore entrance blockade mechanism, and provide an ion channel target in interpreting the pharmacological effects of two commonly used phosphoinositide kinase inhibitors.

[1] Department of Anesthesiology & Perioperative Medicine, The University of Texas MD Anderson Cancer Center, Houston, TX, USA. [2] Neuroscience and Biochemistry & Cell Biology Programs, The University of Texas MD Anderson Cancer Center UTHealth Graduate School of Biomedical Sciences, Houston, TX, USA. ✉email: jyan1@mdanderson.org

Two-pore channels (TPCs) are mainly found in acidic organelles of endolysosomes in animals and also vacuoles in plants. Humans and mice have two functional TPC isoforms: TPC1, which is broadly expressed in different stages of endosomes and lysosomes, and TPC2, which is expressed mainly in late-stage endosomes and lysosomes[1,2]. TPCs are homodimeric cation channels. Each subunit contains two transmembrane domains of the basic structural unit (six transmembrane segments and a pore loop) of a voltage-gated ion channel. TPCs are potently activated by phosphatidylinositol 3,5-bisphosphate $(PI(3,5)P_2)$[3–6], inhibited by ATP via mTORC1[7], and slightly blockaded by cytoplasmic and luminal $Mg^{2+}$[5]. Human TPC1 is voltage-gated and regulated by cytoplasmic and luminal $Ca^{2+}$[8], whereas human TPC2 is not sensitive to voltage or $Ca^{2+}$[5]. The recently reported cryo-electron microscopic (Cryo-EM) structures of mouse TPC1[9] and human TPC2[10] have provided a structural basis for understanding TPC function.

The endolysosomal TPCs regulate the function of the endolysosomal system, including endomembrane dynamics and $Ca^{2+}$ homeostasis of the acidic stores[3,11]. Accumulating evidence supports that TPCs are critical to NAADP-evoked $Ca^{2+}$ release from acidic stores[1,12–18]. TPCs are involved in many cellular processes, including autophagy[19], migration and proliferation of cancer cells[20,21], muscle cell differentiation[22] and contraction[16], and fertilization[23], and they are implicated in pigmentation[24–26], Parkinson's disease[27], and fatty liver disease[28,29]. TPCs are also important to infection mechanisms of viruses, such as Ebola[30], MERS[31], and SARS-COV-2[32].

Potent and/or selective modulators are important pharmacological tools in understanding molecular mechanisms and physiological and pathological function of an ion channel. Currently, the availability of potent and/or selective antagonists of TPCs remains limited. Ned-19 (trans-Ned 19) is a commonly used as an NAADP signaling antagonist[33], albeit it also blocked $PI(3,5)P_2$-induced TPC current[30] and formed complex with plant TPC1 in solved X-ray structure[34]. The potency of Ned-19 in TPC inhibition remains unclear as only a high concentration (200 μM) of Ned-19 was reported to be associated with a significant inhibition (~75%) lysosomal TPC2 activity[30]. Naringenin, a modulator of multiple ion channels, can inhibit TPCs when applied at high concentrations (an $IC_{50}$ of ~200 μM)[35], likely via the blockade of the channel pore. Tetrandrine, a voltage-gated $Ca^{2+}$ $(Ca_V)$ channel blocker, was reported to inhibit Ebola virus entry into host cells presumably via inhibition of TPCs[30]. However, tetrandrine is hardly an optimal TPC antagonist because of its issue in specificity and currently the lack of reported full inhibition of lysosomal TPC2 currents, e.g., 50–60% inhibition by 0.5 μM[30]

and 10 μM tetrandrine[21], in spite of its potent effect on virus entry ($IC_{50}$ of 55 nM)[30]. SG-094, a chemical derivative of tetrandrine, was recently developed to have some improved inhibitory effect (~75% by 10 μM) on TPC2[21]. MT-8, a flavonoid compound isolated from plant extracts, was recently identified to inhibit lysosomal TPC2 effectively with an $IC_{50}$ of 2.6 μM[36]. Some other $Ca_V$ and $Na_V$ channel antagonists, such as nifedipine and lidocaine, were also found to inhibit NAADP-evoked $Ca^{2+}$ elevation in cells and had been proposed to act as antagonists of TPCs[37]. But, electrophysiological evidence for TPC inhibition by these $Ca_V$ and $Na_V$ antagonists is lacking.

We considered two candidates. YM201636 is a potent and selective inhibitor (an $IC_{50}$ of ~30 nM) of PIKfyve, the principal phosphoinositide kinase that produces $PI(3,5)P_2$ via PtdIns3Pphosphorylation[38,39]. Given that $PI(3,5)P_2$ is a key component and regulator of the endolysosomal system, YM201636 is widely used in research studies to disrupt endomembrane transport, e.g., to prevent infection by Zaire ebolavirus and SARS-COV-2[32,40] or to inhibit retroviral release from infected cells[41]. Similarly, PI-103 is a potent multi-target inhibitor of class I phosphatidylinositol 3-kinase (PI3K), mammalian target of rapamycin complex (mTOR), and DNA-dependent protein kinase (DNA-PK)[42]. PI-103 has nearly the same chemical structure as YM201636 but without YM201636's 6-aminonicotinamide group. In this study, we identified YM201636 and PI-103 as potent inhibitors of human TPC2 channels, and we further investigated and identified the mechanisms underlying their inhibitory effects on TPCs.

## Results

### YM201636 suppressed NAADP-evoked $Ca^{2+}$ release and TPC2 activation

TPC channel activities are dually modulated by two endogenous signaling molecules: NAADP and $PI(3,5)P_2$. We tested whether suppression of $PI(3,5)P_2$ production by application of a PIKfyve inhibitor, YM201636, can affect NAADP-evoked $Ca^{2+}$ release. We observed that direct microinjection of YM201636 (1 μM) together with NAADP led to a great reduction of the NAADP-evoked $Ca^{2+}$ elevation by 80% in HEK293 cells (Fig. 1a, b). However, the cells' response to NAADP was largely unaffected upon microinjection of apilimod, another potent PIKfyve inhibitor (Fig. 1a, b). Taking advantage of our recently reported method of measuring NAADP-evoked TPC2 activation[18] using the plasma membrane–targeted TPC2 L11A/ L12A mutant (TPC2$^{PM}$) channels[9,10], we examined the inhibitory effect of YM201636 on NAADP-induced TPC2$^{PM}$ currents in whole cell recording. We recorded the NAADP microinjection-

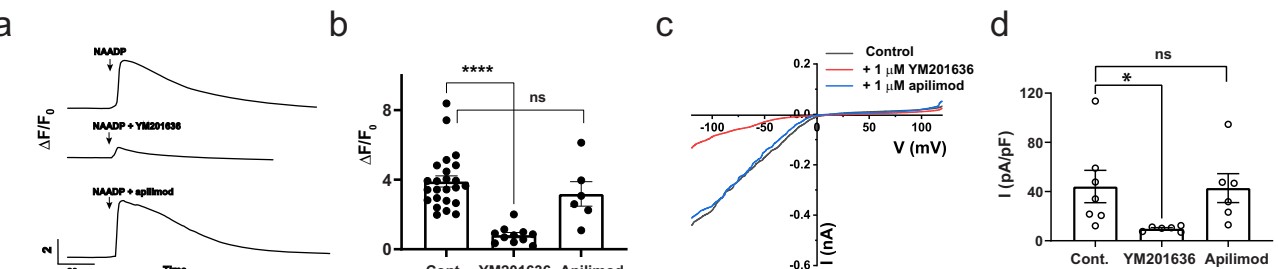

**Fig. 1 YM201636 inhibits NAADP-evoked $Ca^{2+}$ release and TPC2 activation. a** Time course of NAADP (microinjection)-induced change in fluorescence of $Ca^{2+}$ indicator in TPC2-expressing HEK293 cells. NAADP (100 nM), YM201636 (1 μM), and apilimod (1 μM) were included in the injection pipette solution and applied inside cells via microinjection. **b** Averaged NAADP-induced changes in $Ca^{2+}$ indicator fluorescence in TPC2-expressing HEK293 cells. **c** Averaged traces of NAADP (1 μM in pipette solution) microinjection-induced whole cell currents in HEK293 cells transfected with TPC2$^{PM}$ mutant construct. YM201636 (1 μM) and apilimod (1 μM) were incubated with cells in bath solution for ~ 10 min before recording. **d** Averaged current density of NAADP microinjection-induced whole cell TPC2$^{PM}$ currents at −120 mV. The averaged data are presented as mean value ± SEM. Unpaired Student's $t$-test (two tailed) was used to calculate $p$ values. ****, *, and ns are for $p$ values ≤ 0.0001, 0.05, and > 0.05, respectively.

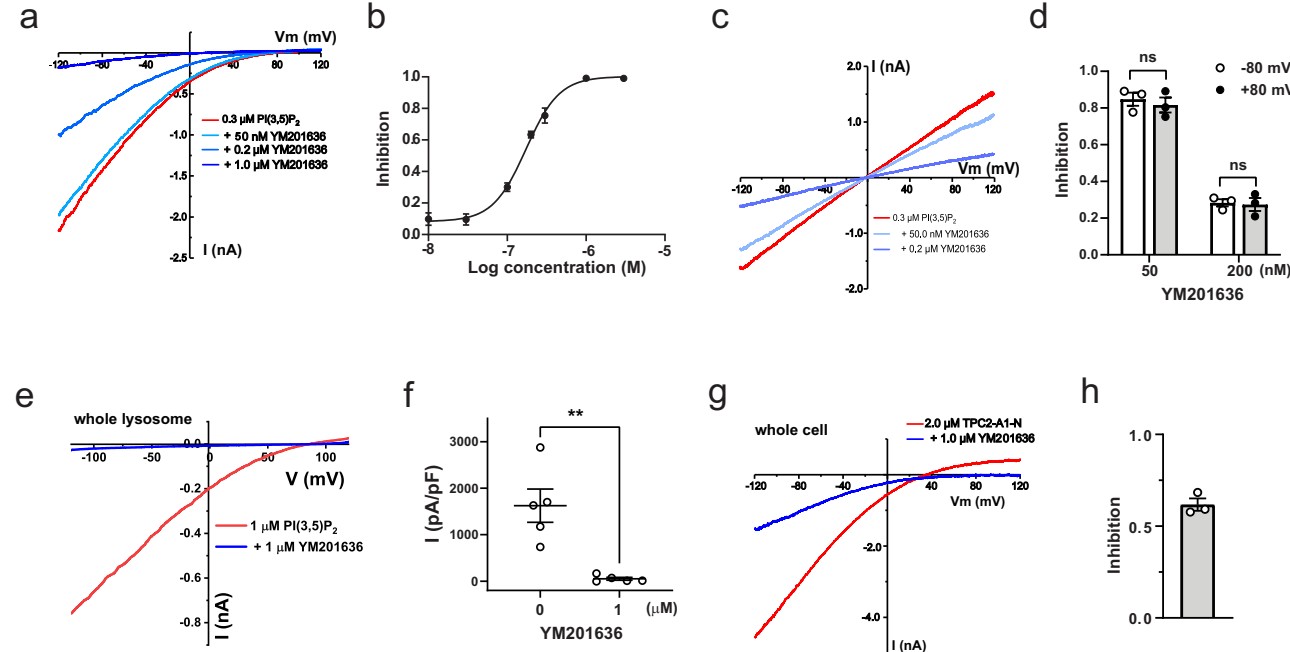

**Fig. 2 YM201636 directly inhibits human TPC2 channels. a, b** Representative current traces (**a**) and plot of the averaged dose response (**b**) of the inhibitory effect of YM201636 on the TPC2$^{PM}$ Na$^+$ currents elicited by 0.3 μM PI(3,5)P$_2$. Path-clamp recording was done in inside-out configuration using asymmetric Na$^+$ (outside)/K$^+$ (inside) recording solutions. The data ($n = 6$–8 for each data point) in (**b**) were fit by a Hill equation. **c** Representative current traces of the TPC2$^{PM}$ Na$^+$ currents recorded in inside-out configuration using symmetric Na$^+$ recording solutions in the absence and presence of 50 nM or 0.2 μM YM201636. **d** Averaged inhibitory effects of YM201636 on the outward and inward TPC2$^{PM}$ currents recorded at +80 and −80 mV as shown in (**c**). **e, f** Representative current traces (**e**) and averaged plot (**f**) of the effects of 1 μM YM201636 on 1 μM PI(3,5)P$_2$-induced TPC2 Na$^+$ currents in whole lysosome patch-clamp recording using asymmetric K$^+$ (cytosolic)/Na$^+$ (lumenal) recording solutions. **g, h** Representative current traces (**g**) and averaged plot at −120 mV (**h**) of the effect of 1 μM YM201636 perfused on the extracellular side on TPC2$^{PM}$ channel currents elicited by extracellularly applied TPC2-A1-N in whole cell patch-clamp recording using asymmetric Na$^+$ (outside)/K$^+$ (inside) recording solutions. The averaged data are presented as mean value ± SEM. Unpaired Student's $t$-test (two tailed) was used to calculate $p$ values. ** and ns are for $p$ values ≤ 0.01 and > 0.05, respectively.

induced TPC2$^{PM}$ currents as we recently reported[18]. The presence of 1 μM YM201636 in the bath solution caused 78% inhibition of the NAADP-induced TPC2$^{PM}$ currents (Fig. 1c, d). Apilimod at 1 μM exerted little effect on NAADP (microinjection)-induced TPC2$^{PM}$ activation (Fig. 1c, d). These results indicate that YM201636 inhibits NAADP-evoked Ca$^{2+}$ release and TPC2 activation in a manner unrelated to PIKfyve because of the lack of effect from apilimod.

**YM201636 directly inhibits TPC2 channels**. We performed inside-out patch-clamp recording of the TPC2$^{PM}$ currents to determine whether YM201636 can directly act on the channel. We observed that YM201636 application from the cytosolic side at sub-micromolar concentrations directly inhibited PI(3,5)P$_2$-induced human TPC2 channel Na$^+$ currents (Fig. 2a). YM201636 potently inhibited TPC2 channels in an antagonist concentration-dependent manner with a low IC$_{50}$ of 0.16 μM (Fig. 2b). YM201636's inhibition of the human TPC2 channel was voltage-independent, as shown by the similar levels of inhibition of outward and inward Na$^+$ currents elicited by PI(3,5)P$_2$ at negative and positive voltages (Fig. 2c, d). To confirm that YM201636 also inhibits the wild type TPC2 expressed on lysosomes, we performed patch clamp recording of whole lysosome (enlarged by vacuolin-1 treatment). As expected, 1 μM YM201636 when applied from the cytosolic side fully abolished the PI(3,5)P$_2$-induced lysosomal TPC2 currents (Fig. 2e, f).

Given the lipophilic property of YM201636, we expected it can also inhibit TPC2$^{PM}$ when applied from extracellular side. To test this, we performed whole-cell recording and applied the inhibitor from the extracellular side, which is analogous to the lumenal side

of the lysosome. In this experiment, we activated the TPC currents by perfusion of a membrane permeable activator TPC2-A1-N[43] on the extracellular side because of the difficulty in manipulation of intracellular application of PI(3,5)P$_2$ in whole cell recording. TPC2-A1-N was considered to be more like NAADP than PI(3,5)P$_2$ in TPC2 activation[43]. We observed that YM201636 at 1 μM caused significant (62 ± 3%, $n = 3$) inhibition of the TPC2-A1-N-induced TPC2$^{PM}$ currents when it was perfused together with the activator from the extracellular side (Fig. 2g, h). The observed reduced inhibition under this condition as compared to when it was applied on the cytosolic side in inside-out configuration likely suggests favorable accessibility of the inhibitory site from the cytosolic side. The concentration of the inhibitor could be lower inside cells than the outside solution, caused by insufficient equilibration of the chemical across membrane during perfusion and also dilution by the pipette solution once the inhibitor is inside the cell.

**The open state-dependence of YM201636's inhibition on TPC2**. PI(3,5)P$_2$, when applied from the cytosolic side, activated human TPC2 channels with an observed activation rate of a $\tau_{on}$ of 9.0 ± 0.5 s ($n = 8$), and the effect could be washed off within 1–2 min with a deactivation rate of a $\tau_{off}$ of 20.8 ± 0.9 s ($n = 7$) (Fig. 3a, d). The time constant $\tau$ is equal to the time taken for a change by a factor of 1- 1/$e$ or ~0.632. We found YM201636 inhibited the human TPC2 channel quickly with a $\tau_{on}$ of 3.4 ± 0.9 s ($n = 5$) when the channels were pre-activated by PI(3,5)P$_2$ (Fig. 3b, d). The wash-off of YM201636 in the presence of PI(3,5)P$_2$, i.e., in the open state, was slow at a $\tau_{off}$ rate of 58.5 ± 3.8 s ($n = 5$) (Fig. 3b, d). To determine the state dependence of inhibitor's action

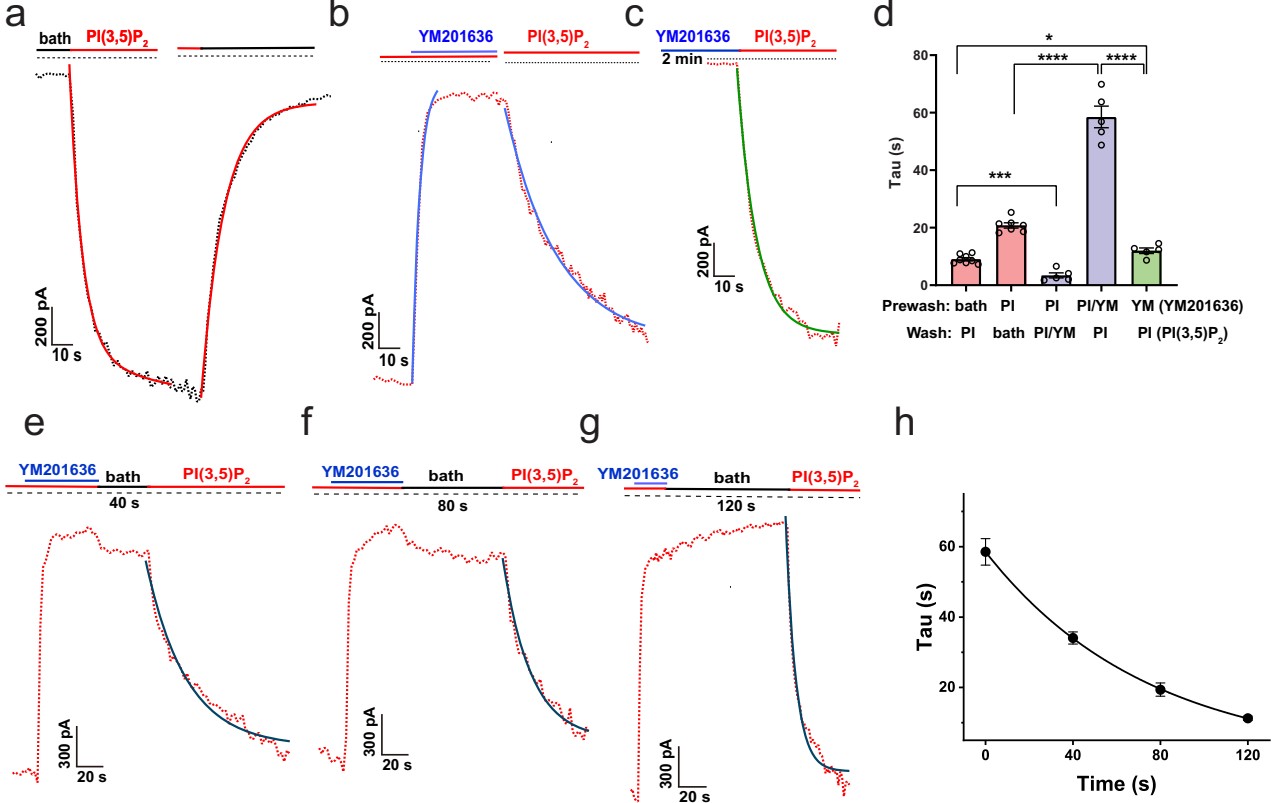

**Fig. 3 The state dependence of human TPC2 inhibition by YM201636. a** Representative TPC2$^{PM}$ currents upon activation by 1.0 μM PI(3,5)P$_2$ and deactivation by wash-off of the ligand. **b** The inhibition and restoration of TPC2$^{PM}$ currents (pre-activated by PI(3,5)P$_2$) upon wash-on and wash-off of 1.0 μM YM201636. **c** The PI(3,5)P$_2$ (1.0 μM)-induced TPC2$^{PM}$ currents after 2-min treatment of the closed channels (in the absence of an activator) with 1.0 μM YM201636. **d** Averaged kinetics (Tau) of TPC2$^{PM}$ currents in the absence and presence of YM201636 and/or PI(3,5)P$_2$ as shown in (**a–c**) ($n = 5$–8). **e–g** The time-dependence of YM201636 wash-off in the absence of an activator. The PI(3,5)P$_2$-actativated TPC2$^{PM}$ was first inhibited by YM201636 and then washed with the bath solution (no activator) for 40 s (e) 80 s (**f**) 120 s (**g**). The residual inhibitory effect of YM201636 was assayed by the slowed kinetics of PI(3,5)P$_2$-induced TPC2$^{PM}$ activation. **h** The averaged kinetics (Tau) of PI(3,5)P$_2$- induced restoration of TPC2$^{PM}$ currents (pre-inhibited by YM201636) after wash-off of the inhibitor by bath solution for different times as shown in (**e–g**) ($n = 5$–8). The 0 s data was taken from (**b**). All currents were acquired at −120 mV in inside-out configuration using asymmetric Na$^+$ (outside)/K$^+$ (inside) recording solutions. The traces were fitted exponentially to obtain the Tau values. The averaged data are presented as mean value ± SEM. Unpaired Student's *t*-test (two tailed) was used to calculate *p* values. ****, ***, * are for *p* values ≤ 0.0001, ≤0.001, and ≤ 0.05, respectively.

on the channel, we applied YM201636 when the channels were in the closed state, i.e., in the absence of PI(3,5)P$_2$ for 2 min, followed by activating the channels by PI(3,5)P$_2$ in the absence of YM201636 (Fig. 3c). We found that the rising rate of TPC2 currents activated by PI(3,5)P$_2$ in the presence YM201636 pre-application remained fast ($\tau_{on} = 12.0 \pm 1.0$ s; $n = 5$) (Fig. 3c, d), which is comparable to that ($\tau = 9.0$ s) in the absence of YM201636 pre-application (Fig. 3a, d) but much faster than the wash-off rate ($\tau = 58.5$ s) of YM201636 (Fig. 3b, d). This suggests that YM201636 barely binds to TPC2 for channel inhibition in the closed state. To determine the state-dependence of YM201636's dissociation from the channel, we evaluated the wash-off rate of YM201636 in the channel's closed state, i.e., in the absence of activator. After the channels were inhibited by YM201636 in the presence of PI(3,5)P$_2$, we washed the excised patches (inside-out) with the bath solution alone (no PI(3,5)P$_2$ and YM201636) on the intracellular side for different time lengths and then applied PI(3,5)P$_2$ to check the residual inhibitory effect of YM201636 on the rate of channel activation by PI(3,5)P$_2$ (Fig. 3e–g). The observed activation rates were $\tau$ (sec) = 34.1 ± 1.7 ($n = 7$), 19.4 ± 1.9 ($n = 5$), and 11.3 ± 0.6 ($n = 6$) after a 40, 80 and 120 s wash with the bath solution, respectively ((Fig. 3e–h). The time course of the wash time dependent increase in the channel activation rate can be fitted with a $\tau$ of 52 s (Fig. 3h), which is close

to the wash-off rate ($\tau = 59$ s) of YM201636 in the presence of PI(3,5)P$_2$ (Fig. 2b, d). Therefore, YM201636 mainly binds to and inhibit the channels when the channels are activated, whereas its dissociation is largely independent of the presence or absence of PI(3,5)P$_2$ or the channel's activation status.

**The ligand-independence of YM201636's inhibition of TPC2.** To determine whether the inhibition of TPC2 by YM201636 is specific to any ligand activation pathway, we managed to generate a constitutively open TPC2$^{PM}$ mutant channel. We performed Ala-substitution mutations in the T308, Y312, L690, and L694 residues (Fig. 4a), which have been predicted from human TPC2 structures to form a bundle-crossing activation gate on the cytosolic side of the channel pore[10]. We were unable to obtain functional channels from the single mutation of L690A or L694A. However, the double mutant channel L690A/L694A produced Na$^+$ currents in the absence of any agonist, and application of PI(3,5)P$_2$ did not increase the currents (Fig. 4b), indicating that the channels are already constitutively fully open. This result provides functional evidence that these two residues are indeed involved in the formation of the activation gate. The T308A and Y312A mutant channels remained closed in the absence of an agonist and were sensitive to PI(3,5)P$_2$ for channel activation, suggesting that these

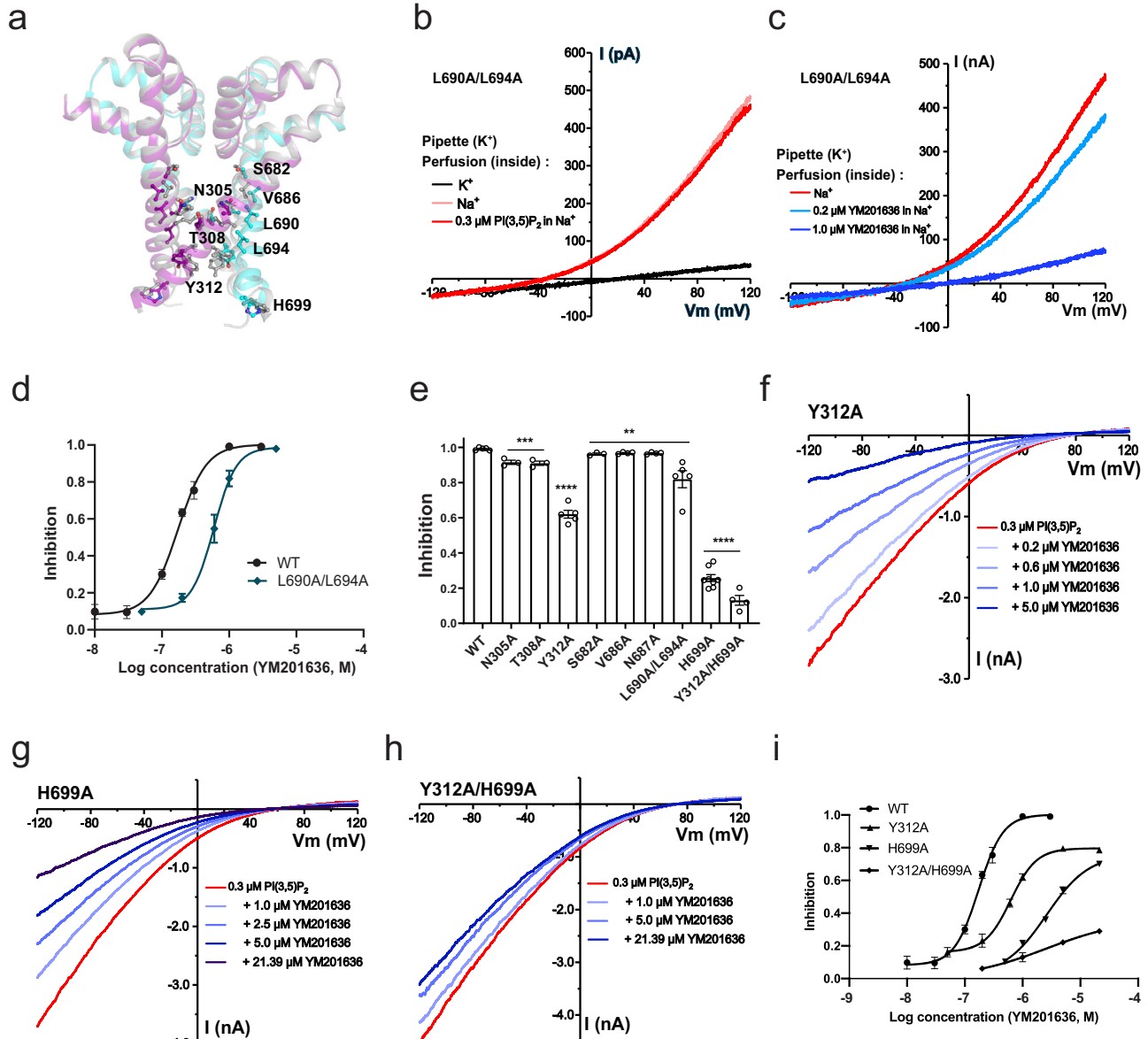

**Fig. 4 The effects of YM201636 on human TPC2 pore region mutants. a** Pore region structures of the PI(3,5)P$_2$-bound open-state (colored) (PDB ID: 6NQ0) and apo/closed-state (gray) (PDB ID: 6QN1) human TPC2 channels showing the positions of the mutated residues in this study. **b** The constitutively-open TPC2$^{PM}$ currents induced by the L690A/L694A double mutation in the absence and presence of PI(3,5)P$_2$. **c** The inhibition of the constitutively-open TPC2$^{PM/L690A/L694A}$ channel currents by YM201636. **d** The dose-response of YM201636's inhibition on the constitutively-open TPC2$^{PM/L690A/L694A}$ channel currents ($n = 4$–6) as compared to that of the WT channels. **e** The averaged inhibitory effects of 1.0 μM YM201636 on TPC2$^{PM}$ pore-region mutant channels. The channels were activated by 0.3 μM PI(3,5)P$_2$. **f–h** The inhibitory effects of YM201636 on human TPC2$^{PM}$ mutants Y312A (**f**), H699A (**g**), and Y312A/H699A (**h**). **i** Dose-responses ($n = 4$–6 for each data point) of the YM201636's inhibitory effect on TPC2$^{PM}$ mutants Y312A, H699A, Y312A/H699A as compared to that of the WT channels. All recordings were done in inside-out configuration using asymmetric Na$^+$ (inside)/K$^+$ (outside) (**b–c**) or Na$^+$ (outside)/K$^+$ (inside) (**e–h**) recording solutions. The averaged data are presented as mean value ± SEM. Unpaired Student's *t*-test (two tailed) was used to calculate *p* values. ****, ***, ** are for *p* values ≤ 0.0001, ≤0.001, and ≤0.01, respectively.

two residues are less important in activation pore-gate formation. With the L690A/L694A mutant channel in the absence of an agonist, we observed that YM201636 could still reduce the channel's constitutively open currents in a concentration-dependent manner, with an IC$_{50}$ of 0.54 μM (Fig. 4c, d). Together with the above observed the YM201636's inhibition of TPC2 activated by PI(3,5)P2, NAADP, or TPC2-A1-N, this result clearly indicates that the YM201636's inhibition of TPC2 is independent of activation pathways. This inhibitory property of YM201636 agrees with a channel pore blocker. The more than 3-fold increase of YM201636's IC$_{50}$ by the L690A/L694A double mutation also

suggests a role of the pore structure in TPC2 inhibition by this inhibitor. Therefore, we consider YM201636 most likely a TPC2 open-channel pore blocker.

**TPC2 inhibition by YM201636 is sensitive to mutations at and near the cytosolic side pore entrance.** To identify the YM201636 binding sites along the channel pore, we performed an Ala scanning analysis among the channel pore residues (Fig. 4a). Given its large size, the YM201636 molecule may interact with residues inside the pore and also those near the pore entrance. Therefore, we also examined the mutational effect of His699

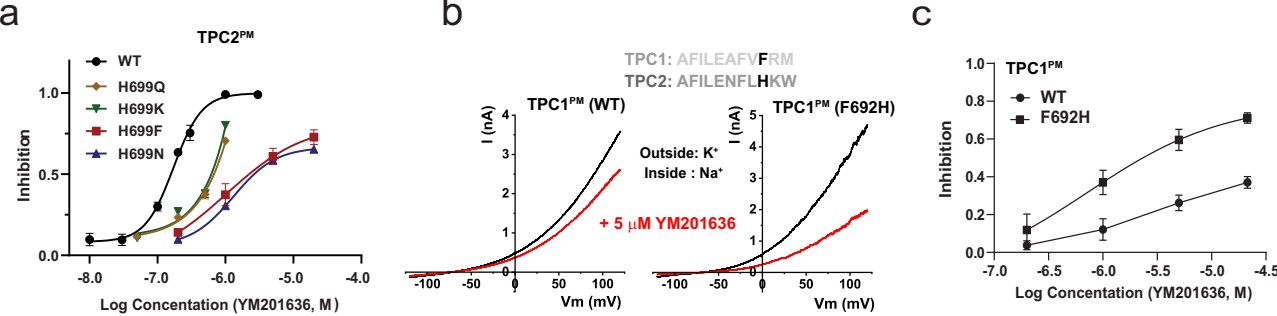

**Fig. 5 The residue H699 underlies the increased sensitivity of TPC2 to YM201636 than TPC1. a** The effect of mutations at H699 on the dose-responses ($n = 5$–$7$ for each data point) of the YM201636's inhibition of the TPC2$^{PM}$ channel. **b** The effect of YM201636 on WT and mutant H692F of human TPC1$^{PM}$ channel. The sequence alignment of human TPC1 and TPC2 at the mutated region was showed on top. Phe692 in TPC1 and His699 in TPC2 channel were indicated in black. **c** Dose-responses ($n = 3$–$8$ for each data point) of YM201636's inhibitory effect on WT and H692F mutant channels of human TPC1$^{PM}$. All recordings were done in inside-out configuration using asymmetric Na$^+$ (outside)/K$^+$ (inside) (**a**) or Na$^+$ (inside)/K$^+$ (outside) (**b**, **c**) recording solutions. The averaged data are presented as mean value ± SEM.

residue, which is located immediately at the pore entrance on the cytosolic side (Fig. 4a). Compared to the L690A/L694A double mutation, the Ala-substitution of residues inside the channel pore by mutations N305A, T308A, S682A, V686A, and N687A showed less effects on TPC2 inhibition by 1 μM YM201636 (Fig. 4e). Major reductions in sensitivity to YM201636 were observed with mutations of the Y312 and H699 residues (Fig. 4e–i). Y312 is located immediately below the L690/L694 pore-gate while H699 is positioned near the cytosolic side of the pore entrance (Fig. 4a). The Y312A mutation significantly increased the IC$_{50}$ for TPC2 inhibition by YM201636, by more than 4-fold (IC$_{50} = 0.67$ μM) (Fig. 4f, i). The H699A mutation drastically reduced the channel's sensitivity to YM201636, as indicated by a more than 20-fold increase in the IC$_{50}$ (IC$_{50} = 4.33$ μM) compared to that of the wild-type channel (Fig. 4g, i). The double mutation Y312A/H699A resulted in a much greater loss of the channel's sensitivity to YM201636, with an IC$_{50}$ beyond the highest tested concentration (21 μM) (Fig. 4h, i), i.e., the mutation resulted in an increase of IC$_{50}$ by more than 100-fold compared to that of the wild-type channel. These results indicate that the cytosolic-side pore-gate forming residues L690, L694, and Y312, together with the H699 residue located immediately outside of the pore, determines YM201636's inhibitory effect on TPC2, likely by forming the inhibitor's binding sites.

**His699 underlies much greater sensitivity to YM201636 in TPC2 than TPC1.** To further analyze the impact of the mutations of H699 on TPC2's sensitivity to YM201636, we generated more mutations at this site. Similar to the effects of the H699A mutation, the H699F and H699N mutations greatly reduced inhibition of the channel by YM201636 with estimated IC$_{50}$ values of ~2.2 and ~2.68 μM, respectively (Fig. 5a). The H699Q and H699K mutations also reduced TPC2's sensitivity to YM201636 but to a much lesser extent (IC$_{50}$ values of ~0.7 μM and ~0.6 μM, respectively) than H699A, H699F, and H699N mutations did (Fig. 5a). These suggest that substitution of histidine by a larger polar (Gln as compared to Asn) or positively charged residue at this position helps alleviate the loss in the channel's sensitivity to YM201636.

Interestingly, the TPC1 channel harbors a Phe (F692) at the equivalent TPC2-H699 position (Fig. 5b). By patch-clamp recording of the plasma membrane-targeted TPC1 L11A/I12A mutant (TPC1$^{PM}$) channel[9,18], we observed that TPC1, as compared to TPC2, was much less sensitive to YM201636 (IC$_{50} > 20$ μM) (Fig. 5b, c). Upon substitution with a histidine at this position by the F692H mutation, the TPC1 channel's

sensitivity to YM201636 was greatly enhanced with a resulting IC$_{50}$ of ~2.3 μM (Fig. 5b, c). Therefore, H699 is a key determinant for TPC2 channel's inhibition by YM201636, whose substitution with a Phe in the TPC1 channel partially accounts for the vastly decreased sensitivity to YM201636.

**PI-103, a YM201636 analog, also acts as a TPC2 pore blocker.** PI-103, a known PI3K and mTOR inhibitor, has nearly the same chemical structure as YM201636 but without a 6-amino-nicotinamide group (Fig. 6a). We observed that PI-103 also directly inhibited the PI(3,5)P$_2$-induced TPC2 channel Na$^+$ current (Fig. 6b) in a concentration-dependent manner with an IC$_{50}$ of 0.64 μM (Fig. 6e), which is 4-fold higher than that of YM201636, suggesting that the 6-amino-nicotinamide group has a role in enhancing the potency of YM201636's TPC2 blockade effect. Similar to that observed with YM201636, the Y312A mutation in TPC2 also resulted in reduced inhibition of the channel by PI-103 with an elevated IC$_{50}$ of 13.9 μM (Fig. 6c, e). The H699A mutation largely abolished the channel's blockade by PI-103 (Fig. 6d, e). Therefore, PI-103 acts similarly to YM201636 by functioning as a potent TPC2 channel blocker.

**Molecular docking and dynamic simulation analyses of YM201636's bindings along the channel pore.** We initially performed a virtual molecular docking analysis via the AutoDock Vina program[44] using the reported human TPC2 Cryo-EM structures[10] directly. With the whole proteins included in the grid space for docking and a cut-off affinity of $-8.5$ kcal/mol, YM201636 molecule docked inside the channel pore was observed in 13 out of 34 poses for the PI(3,5)P$_2$-bound open-state structure (PBD ID:6NQ0), 3 out of 22 poses for the PI(3,5)P$_2$-bound closed-state structure (PBD ID:6NQ2), and none out of 7 for apo/closed-state structure (PBD ID:6NQ1). This result is consistent with our finding of the requirement of the channel's open-state for channel inhibition by YM201636. Thus, we focused on docking analysis of YM201636's bindings on the PI(3,5)P$_2$-bound open-state structure.

Most molecular docking programs including AutoDock Vina treat the receptor proteins as rigid bodies to be computation efficient but at a cost of limitation in accuracy because of the dynamic nature of protein-ligand binding involving the protein's local conformational changes in binding. To better identify the inhibitors' binding sites in the TPC2 open structure, we first performed a molecular dynamic simulation of the PI(3,5)P$_2$-bound human TPC2 open-state structure (PDB: 6NQ0)[10]. After simulation for 200 ns, we clustered the trajectory structural frames from the last 50 ns of the simulation and generated 76 representative snapshots

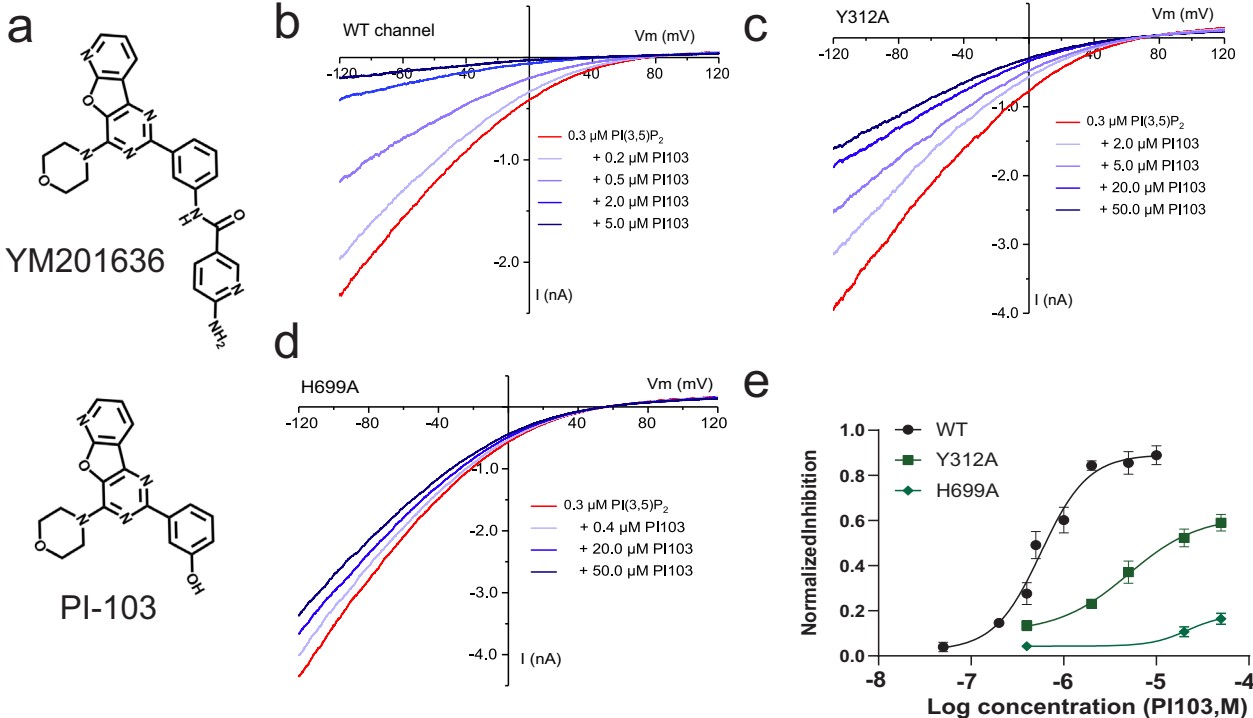

**Fig. 6 Inhibition of the human TPC2 channel by PI-103. a** The chemical structures of YM201636 and PI-103. **b–d** The effect of PI-103 on WT (**b**) and Y312A (**c**) and H699A (**d**) mutants of the human TPC2$^{PM}$ channel. The channel was activated by 0.3 μM PI(3,5)P$_2$. **e** Dose-responses ($n = 5$–7 for each data point) of the PI-103's inhibitory effect on WT and Y312A and H699A mutants of the human TPC2 channel, as shown in (**b–d**). All recordings were done in inside-out configuration using asymmetric Na$^+$ (outside)/K$^+$ (inside) recording solutions. The averaged data are presented as mean value ± SEM.

of the simulated dynamic structures. With them, we performed virtual molecular docking analyses individually via the AutoDock Vina program and allowed the output of the top 20 poses based on calculated affinity. Among the ~1500 generated poses, we chose the 30 poses with the highest affinity for visual examination and further analysis. If the whole region of the channel pore was included in the grid space for docking, the top 30 poses had an average affinity of −11.05 kcal/mol, and all YM201636 molecules were docked at the bundle-crossing pore-gate region flanked by the Y312, R316, L690, L694, and E695 residues from the two identical subunits (Fig. 7a). No preference in the orientation of the YM201636 molecule was observed, as the 6-amino-nicotinamide group pointed in and out of the pore in about equally often. PI-103 was similarly docked at a similar region, at which the top 30 poses had an average affinity of −10.11 kcal/mol, which was slightly lower in affinity than that of YM201636 and consistent with the increased IC$_{50}$ of PI-103 compared to that of YM201636. However, the interactions of YM201636 and PI-103 with H699 were very limited or absent in their top 30 energetically favorable poses as the inhibitors were well-docked inside the pore whereas H699 is located outside of the pore.

Given the importance of the H699 residue in TPCs' sensitivity to the inhibitors, we hypothesized that H699 potentiates the channel blockade by interactions with the inhibitors around the pore entrance, allowing the inhibitor to block ion conductance directly at the pore entrance and/or alternatively the H699 to serve as initial docking sites to guide and facilitate the inhibitors to move toward the more favorable binding sites inside the pore. To identify the inhibitors' binding poses around the pore entrance, we limited the docking grid space to the pore entrance region. With this docking space restriction, we were able to observe YM201636's bindings at the pore entrance region below the pore-gate with a suboptimal average affinity of −9.27 kcal/mol for the top 30 poses (Fig. 7b). Among most ($n = 26$) of these 30 top poses, the imidazole ring of H699 interacted closely

(within 4 Å excluding hydrogen atoms) with YM201636. The interactions with H699 appeared flexible, involving nearly all the different parts of YM201636 in different poses, suggesting that these interactions are likely dynamic. Similarly, PI-103 was also observed to bind at the pore entrance, although the averaged affinity for the top 30 poses was reduced to −8.19 kcal/mol.

We selected 4 representative poses of YM201636 in complex with TPC2 (Fig. 7c) for further analysis by molecular dynamic simulations for 100 ns. For the first three simulations, the initial poses appeared to be only relative stable for only a short period, e.g., ~20 ns, and then became more mobile and adopted binding modes different from the initial one. However, no full escape from the pore entrance was observed during these 100 ns simulations. For the fourth simulation, the YM201636's interactions with the channel pore became enhanced in that the inhibitor moved slightly inward the pore and kept the pore entrance blocked during the 100 ns simulation as indicated by some interactions with the pore gate region residues L694 and A309 (Fig. 7c). With the PyContact program[45], we performed a systematic analysis of the interactions between YM201636 and protein residues in the four 100 ns simulation trajectories and found that YM201636 remained strongly interacting with H699 most time (Fig. 7d), including H-bond interaction (20% in average). According to the mean contact score and total contact time, YM201636 interacted predominantly with R316, H699, and E695, secondarily with L698 and M320, and marginally with Y312, S313, and N696 (Fig. 7d). The results of molecular dynamic simulation on YM201636's bindings at and near the pore entrance indicate that the inhibitor dynamically interacts with the residues around the pore entrance and can move inside the pore for more sustained channel blockade. Similar to H699, we expect that R316 and E695 also play an important role in TPC2 inhibition by YM201636. The equivalent residues of R316 and E695 in *X. tropicalis* TPC3 were found to be important for

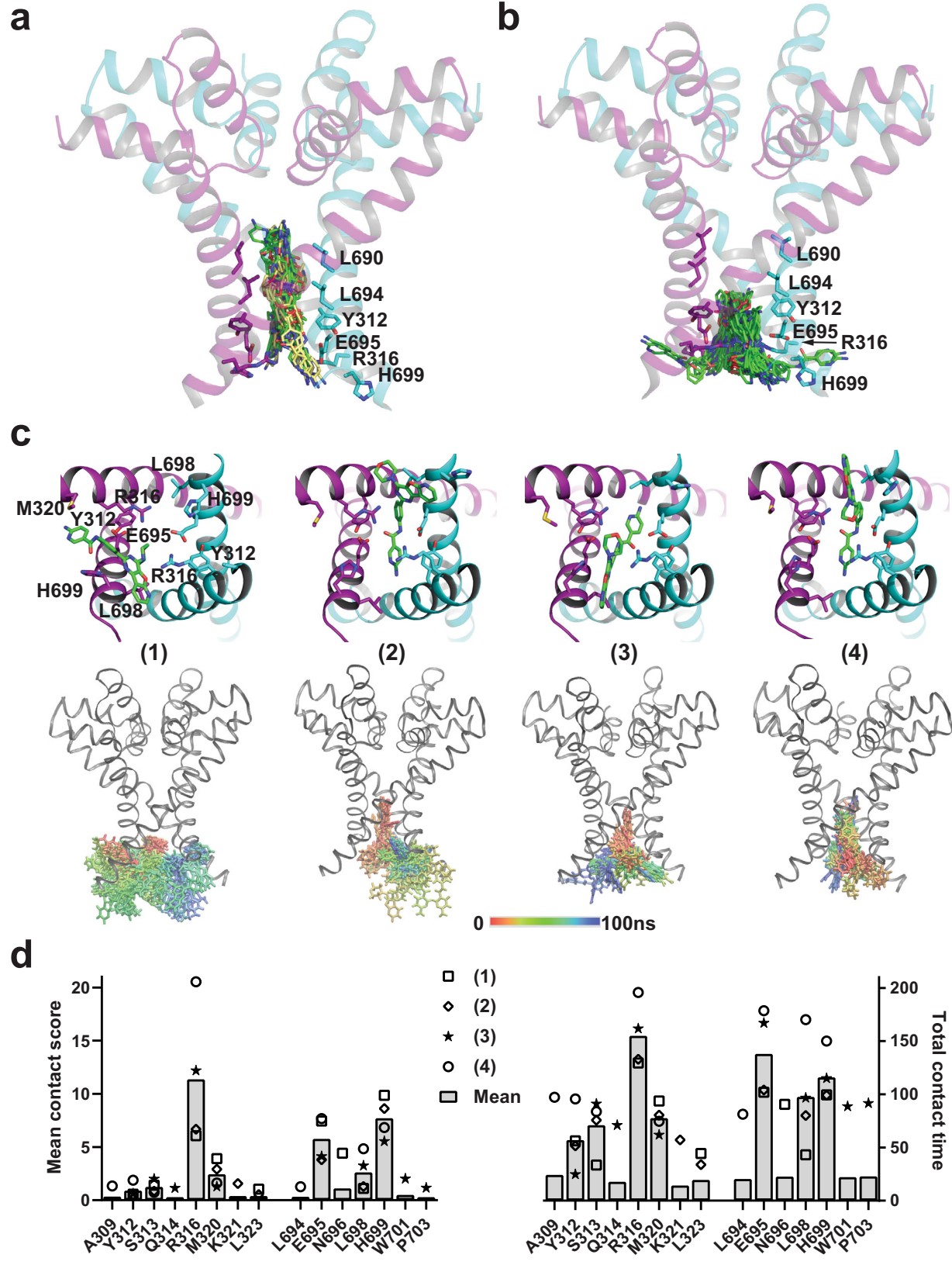

the channel gating via electrostatic interactions and their mutations could result in non-functional channels[46]. Because both R316A and E695A mutations in human TPC2 produced non-functional channels, we pursued no further mutational analysis on them.

## Discussion

In this study, we identified YM201636 and its analog PI-103 to be potent human TPC2 channel blockers. YM201636 and PI-103 act similarly on TPC2 as their inhibitory effects on the channels are similarly affected by pore mutations. Importantly, as pore

**Fig. 7 Molecular docking and molecular dynamic simulation analyses of the YM201636's bindings in human TPC2. a, b** The top 30 poses of YM201636 docked in the whole channel pore region (**a**) or the pore entrance region (**b**) of the PI(3,5)$P_2$-bound open-state channels. The YM201636's carbon atom in (**a**) is shown in green and yellow, respectively, for the poses with their 6-amino-nicotinamide group pointed in and out of the pore. **c** The bottom views of the four representative poses (upper panels) and the side-views of the superimposed YM201636's conformations (50 frames; 2 ns/frame) during the 100 ns simulation (bottom). For clarity, only a representative protein structure from a single frame is shown in (**a**–**c**). **d** Plots of the mean contact scores and total contact times for residues interacting with YM201636. The data were obtained by analyses of their interactions in the trajectories of the four 100 ns molecular dynamic simulations with the Pycontact program. The scores and time were combined from the two identical residues of the two homodimeric subunits.

blockers, YM201636 and PI-103 can block the channels in an agonist-independent manner as YM201636 inhibited NAADP-evoked $Ca^{2+}$ elevation, and both NAADP and PI(3,5)$P_2$-activated TPC2 currents, and mutation-induced constitutively open TPC2 channels. Because of their submicromolar $IC_{50}$ values, we considered YM201636 and PI-103 the most potent TPC2-selective antagonists identified thus far. YM201636 and PI-103 are TPC2 selective, as they are much less effective on TPC1 largely because of the His ↔ Phe switch near the cytosolic-side pore entrance (H699 on TPC2 vs. F692 on TPC1). The His ↔ Phe switch between TPC2 and TPC1 is a conserved feature in most animals except in some species, such as *D. rerio* and *S. purpuratus* whose TPC2 has a Tyr and Thr at this position, respectively.

We explored the mechanism of TPC2 inhibition by YM201636 and PI-103. First, we found YM201636 acts only when the channel is in an open state, i.e., pretreatment in the closed state has no effect. However, its inhibitory effect is independent of the mechanisms of channel activation, consistent with the property of an open-channel pore blocker. Our mutational analyses showed the importance of the L690, L694, Y312, and H699 residues located at or near the cytosolic end of the channel pore on TPC2 inhibition by YM201636 and PI-103. The role of Y312 can be easily understood as it sits at the very cytosolic end of the channel pore and its side-chain forms the port to the channel pore. Similarly, the L690A/L694A double mutation, which caused the channel to constitutively open, alters the channel pore structure and thus affects the inhibitors' potency. However, structural perturbation of the H699 residue, which sits immediately outside of the pore, produced the largest impact on the TPC inhibition by YM201636 and PI-103. Its substitution with a Phe in human TPC1 also largely accounts for the greatly reduced sensitivity to the inhibitor. The role of H699 in TPC2 inhibition by YM201636 appears to be indispensable as mutations to other amino acids, regardless of size, polarity, or charge, all resulted in a loss of the channel's sensitivity to the inhibitor to some extent. The histidine residue plays a unique role in protein structure and function. Its imidazole side chain gives rise to its unique aromaticity and acid/base properties at a physiologic pH. YM201636 and PI-103 are chemicals of multiple rings with both aromatic and some polar properties. H699 could interact with YM201636 and PI-103 via both Van der Waals and hydrogen bond interactions including the π-π stacking, cation-π (if histidine is protonated), and hydrogen-π interactions[47]. Our molecular dynamic simulation analysis suggests that H699 can interact with YM201636 and PI-103 and contribute to the inhibitors' initial docking around the pore entrance. Overall, our data favor the possibility that YM201636 binding and blockade mainly occur at the cytosolic end of the pore as mutations deep in the pore had much less effect, and the H699 residue, which is immediately outside the pore, plays a key role in inhibition by interactions with the inhibitors around the pore entrance, allowing the inhibitor to block ion conductance directly at the pore entrance and/or alternatively serve as initial docking sites to facilitate the inhibitors to bind inside the pore. The slower process of YM201636's wash-off than its wash-on agrees with the notion that the inhibitor initially binds and blocks the channel at

or near the pore entrance and then moves more inside the pore for more sustained channel blockade.

Both YM201636 and PI-103 have been widely used in research to target other proteins. Our studies thus identify an important protein target of these two drugs. This also raises caution in interpretation of the potential mechanisms underlying the pharmacological effects of these two drugs, as the blockade of TPC2 could result in a broad range of cellular, physiological, and pathological effects as well. For example, PI-103 has some anti-tumor activity[48,49], and TPC2 is also considered to be implicated in cancer[20]. YM201636 has anti-viral activity[32,40], and TPC2 also matters for virus entry[30,32]. YM201636 is mainly used in research to target PIKfyve and block PI(3,5)$P_2$ production. Although TPC2 is an effector of PI(3,5)$P_2$ signaling, it can also be activated by other mechanisms, e.g., by NAADP via Lsm12 for TPC-mediated $Ca^{2+}$ mobilization[18]. Therefore, direct blockade of TPC2 by YM201636 can have a more profound effect than that caused by a reduction in PI(3,5)$P_2$ synthesis via inhibition of PIKfyve activity. YM201636 and its derivative-based therapeutics could be an effective strategy to simultaneously target two virus entry-related proteins, PIKfyve and TPC2.

Given their broad physiological and pathological roles, TPCs are emerging as important therapeutic targets for many diseases including COVID-19. Currently, there is an unmet need to develop specific and potent antagonists targeting TPCs. Our identification of YM201636 and PI-103 as potent TPC2-selective (over TPC1) blockers and revelation of the underlying mechanism provide effective pharmacological tools to inhibit TPC2 currents and offers templates for rational design of specific and potent inhibitors of TPC2.

## Methods

**Cell culture, plasmids, and transfection**. HEK293 cells were cultured in Dulbecco's modified Eagle's medium with 10% fetal bovine serum, 1% penicillin, and streptomycin in a 5% $CO_2$ incubator. Similar to our recent report[18], recombinant cDNA constructs of human TPC1 (GenBank: AY083666.1) and human TPC2 (GenBank: BC063008.1) with FLAG and V5 epitopes on their C-termini were constructed with pCDNA6 vector (Invitrogen). To facilitate identification of transfected cells, an IRES-containing bicistronic vector, pCDNA6-TPC2-V5-IRES-AcGFP[18], was used in the electrophysiological experiments. Mutations were made with QuikChange II XL Site-Directed Mutagenesis Kit (Agilent Technologies). Cells were transiently transfected with plasmids with transfection reagent of Lipofectamine 2000 (Invitrogen) or polyethylenimine "Max" (PEI Max from Polysciences) and subjected to experiments within 16–48 h after transfection. For the cell health of the mutant L690A/L694A after transfection, 2.0 μM YM201636 was added into the complete serum medium to block TPC2. For human TPC1 channels, pEGFP-C1 was cotransfected at the same time to identify transfected cells for patch clamp recording. Cells were treated with 1% trypsin 4–6 h after transfection and seeded on polylysine-treated glass coverslips soaked in an incubator until recording.

**Imaging analysis of NAADP-evoked $Ca^{2+}$ release**. $Ca^{2+}$ imaging analysis of NAADP-evoked $Ca^{2+}$ elevation was performed as we recently described[18]. Briefly, cells were co-transfected with cDNA constructs of human TPC2 and the $Ca^{2+}$ reporter GCaMP6f, and the transfected cells were identified by GCaMP6f fluorescence. Fluorescence was monitored with an Axio Observer A1 microscope equipped with an AxioCam MRm digital camera and ZEN Blue 2 software containing a physiology module (Carl Zeiss) at a sampling frequency of 2 Hz. Cell injection was performed with a FemtoJet microinjector (Eppendorf). The pipette solution contained 110 mM KCl, 10 mM NaCl, and 20 mM Hepes (pH 7.2) supplemented with

Dextran (10,000 MW)-Texas Red (0.3 mg/ml) and NAADP (100 nM) or vehicle. When needed, 10 μM YM201626 or apilimod at was added to the pipette solution. The bath was Hank's balanced salt solution, which contained 137 mM NaCl, 5.4 mM KCl, 0.25 mM $Na_2HPO_4$, 0.44 mM $KH_2PO_4$, 1 mM $MgSO_4$, 1 mM $MgCl_2$, 10 mM glucose, and 10 mM HEPES (pH 7.4). To minimize interference by contaminated $Ca^{2+}$, the pipette solution was always treated with Chelex 100 resin (#C709, Sigma-Aldrich) immediately before use. Microinjection (0.5 s at 150 hPa) was made ~30 s after pipette tip insertion into cells. Only cells that showed no response to mechanical puncture, i.e., no change in GCaMP6f fluorescence for ~30 s, were chosen for pipette solution injection. Successful injection was verified by fluorescence of the co-injected Texas Red. Elevation in intracellular $Ca^{2+}$ concentration was reported by a change in fluorescence intensity measured as $\Delta F/F_0$, calculated from NAADP microinjection-induced maximal changes in fluorescence ($\Delta F$ at the peak) divided by the fluorescence immediately before microinjection ($F_0$).

**Electrophysiology**. For inside-out and whole cell patch-clamp recording, HEK293 cells were transiently transfected with plasma membrane–targeted $TPC2^{L11A/L12A}$ ($TPC2^{PM}$) or $TPC1^{L11A/I12A}$ ($TPC1^{PM}$) mutant channels using the transfection reagent of PEI MAX as we did before[18]. After 24 h of transfection, human $TPC1^{PM}$ or $TPC2^{PM}$ channel currents were acquired at room temperature using an EPC-10 amplifier and PatchMaster software (HEKA) or a MultiClamp 700B amplifier and pCLAMP software (Axon Instruments). For most inside-out recording of excised plasma membrane patches, the bath solution contained 145 mM $KMeSO_3$, 5 mM NaCl, and 20 mM HEPES (pH 7.35), and the pipette solution contained 145 mM $NaMeSO_3$, 5 mM NaCl, and 10 mM HEPES (pH 7.35). For the constitutively open L690A/L694A mutant TPC2 or human TPC1 channel, the solutions were switched, i.e., the $K^+$-based solution was used as the pipette solution, and the $Na^+$-based solution was used in the bath instead. To measure the voltage dependence of TPC2 inhibition by YM201636, the same $Na^+$-based solution was used on both sides (pipette and bath). Patch pipettes were polished with a resistance of 2–3 MΩ for recording. Similarly, for whole-cell recording to allow inhibitor application on the extracellular side, the $K^+$-based solution was used as the pipette solution, and the $Na^+$-based solution was used in the bath. The TPC2 and TPC1 channel currents were elicited by perfusion of PtdIns(3,5)$P_2$ diC8 (#P-3058, Echelon) on the intracellular side in inside-out recording or by perfusion of TPC2-A1-N (MedChemExpress) on the extracellular side in whole-cell recording with a voltage ramp protocol of −120 mV to +120 mV over 200 ms for every 2 s. YM201636 was applied together with the activator by perfusion.

Whole cell patch-clamp recording of the NAADP (microinjection)-induced $TPC2^{PM}$ currents was performed as we reported[18]. Bath solution contained 145 mM $NaMeSO_3$, 5 mM NaCl, and 10 mM HEPES (pH 7.2). Pipette electrodes (3–5 MΩ) were filled with 145 mM $KMeSO_3$, 5 mM KCl, and 10 mM HEPES (pH 7.2). The cells were visualized under an infrared differential interference contrast optics microscope (Zeiss). Currents were recorded by voltage ramps from −120 to +120 mV over 400 ms for every 2 s with a holding potential of 0 mV. After a whole cell recording configuration was achieved, an injection pipette was inserted into the cell and the baseline of the whole cell current was recorded. Microinjection of NAADP was performed as above in imaging analysis of NAADP-evoked $Ca^{2+}$ release. The NAADP-induced currents were obtained by subtraction of the baseline from NAADP injection-induced currents. YM201636 and apilimod at 1 μM were added in the bath solution for ~10 min before recording.

Whole lysosome patch-clamp recording of PI(3,5)$P_2$-activated TPC2 activation was performed as previously reported by others[43,50] and us[18]. Cells were treated with vacuolin-1 (1 μM) overnight to enlarge endolysosomes. Patch pipettes for recording were polished and had a resistance of 5–8 MΩ. The cytoplasmic solution contained 145 mM $KMeSO_3$, 4 mM NaCl 4, 0.39 mM $CaCl_2$, 1 mM EGTA, and 10 mM HEPES (pH 7.2) (pH was adjusted with KOH). The luminal solution contained 140 mM $NaMeSO_3$, 5 mM $KMeSO_3$, 2 mM Ca($MeSO_3$)$_2$, 1 mM $CaCl_2$, 10 mM HEPES and 10 mM MES (pH 4.6) (pH was adjusted with methanesulfonic (MeSO$_3$) acid). YM201636 was applied together with PI(3,5)$P_2$ on the cytosolic side by perfusion.

All reagents were purchased commercially: PI-103 (#1728; Biovision), YM201636 (#sc-204193; Santa Cruz Biotechnology), apilimod (#sc-480051; Santa Cruz Biotechnology). Dose curves were fitted by the Hill logistic equation. $\tau_{on}$ and $\tau_{off}$ were acquired from singe exponential fitting.

**Molecular docking analysis and molecular dynamic simulation**. Molecular docking analyses of the bindings of the inhibitor YM201636 on TPC2 channel structures were performed using AutoDock Vina program[44] according to the developers' instructions with Cryo-EM structures (PDB IDs: 6NQ0, 6NQ2, and 6NQ1) of human TPC2[10] either directly or after molecular dynamic simulation in the closed state in the presence of and the absence of PI(3,5)$P_2$ (PDB ID: 6NQ2 and 6NQ1)[10]. For molecular dynamic simulation, the Cryo-EM structure of human TPC2 in the open state in complex with PI(3,5)$P_2$ (PDB ID: 6NQ0) was used. The missed flexible C-terminus (residues 702-752) in the original structure was added by modeling with the GalaxyFill algorithm[51] integrated in the CHARMM-GUI webserver[52]. The protein/lipid/solvent systems and input files for molecular dynamic simulation were generated with the CHARMM-GUI webserver[52]. The structural model was embedded in a lipid bilayer of 1-palmitoyl-2-oleoyl-sn-glycero-3-phosphocholine (POPC) within a water box containing 0.15 M KCl in which the protein charges were neutralized with $K^+$ or $Cl^-$ ions. The molecular

dynamic simulation was carried out with Gromacs 2021 (https://doi.org/10.5281/zenodo.5053220)[53] and the CHARMM36m force-field[54] with the WYF parameter for cation-pi interactions[55]. The system was energy-minimized and then equilibrated in 6 steps using default input scripts for Gromacs generated by the CHARMM-GUI webserver. After the equilibration, the systems were simulated for 200 ns with a 2 fs time step. The Nose-Hoover thermostat and a Parrinello-Rahman semi-isotropic pressure control were used to keep the temperature at 303.15 K and the pressure at 1 bar, respectively. A 12-Å cut-off was used to calculate the short-range electrostatic interactions, and the Particle Mesh Ewald summation method was employed to account for the long-range electrostatic interactions.

*Statistics and reproducibility*. The data were processed and plotted with Igor Pro (v5), GraphPad Prism (v9), or OriginLab (v2015 or 2017). All statistical values are performed as means ± standard errors of the mean of *n* repeats of the experiments. Unpaired Student's *t*-test (two-tailed) was used to calculate *p* values. Unless indicated, all measurements or repeats were taken with distinct samples or cells. Independent experiments with similar results related to representative results were done ≥3 times.

**Reporting summary**. Further information on research design is available in the Nature Research Reporting Summary linked to this article.

## Data availability
All relevant data are contained within this article. Source data are found in Supplementary Data. All other data are available from the corresponding author on reasonable request.

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

## Acknowledgements

We thank Ashli R. Villarreal and Sarah Bronson at Research Medical Library of MD Anderson Cancer Center for editing this article. This work was supported by National Institutes of Health grants GM130814 (J.Y.).

## Author contributions

C.D. performed most electrophysiological experiments. X.G. performed calcium imaging experiment and patch-clamp recording of whole lysosomal and whole cell (NAADP-induced) TPC currents. J.Y. performed molecular docking and molecular dynamic simulation analyses. C.D., X.G., and J.Y. designed experiments, analyzed data, and wrote the manuscript.

## Competing interests

The authors declare no competing interests.
