## [Peer Review File · Communications Biology]

Reviewers' comments:

Reviewer #1 (Remarks to the Author):

The manuscript reports TPC blockers and their action mechanisms. The authors accidentally found that YM201636 and apilimod are different in inducing HEK293 cells' response to NAADP. Then using inside-out patch-clamp recording, they revealed that YM201636 directly acted on TPC2. The authors further suggested that YM201636 is a TPC2 open-channel pore blocker and revealed residues critical for TM201636 binding. H699 is a determinant for the greater sensitivity to YM201636 in TPC2 than TPC1. The study also showed that PI-103, a YM201636 analog, also is a TPC2 blocker. Lastly, computations examined the action mechanism of YM201636 on the channel pore. Although I think that the finding of PI-103 does not contribute significantly to the study and it is better to report an analog active to TPC2 whereas inactive to phosphoinositide kinase, I still recognize the manuscript as a systematic and nice work. And the manuscript has been well written. I recommend to publish the paper as it is.

Reviewer #2 (Remarks to the Author):

This is an excellent and convincing contribution from the lab of Jiusheng Yan on the pharmacology of mammalian two-pore-channels (TPCs). The study is largely an electrophysiological one investigating the effects of drugs proposed to block these channels and the important discovery that PIKfyve inhibitors are potent open-state inhibitors of TPC2. This latter point is particularly important since these inhibitors are widely employed to reduce the synthesis of PI35P2, an endolysosomal inositol lipid, are implicated as potential anti-cancer and anti-viral drugs, and a modulator itself of endolysosomal cation channels including TPCs.

The approach used is to use mutated channels that target to the plasma membrane rather than their natural loci in the endolysosomal system, thus making their electrophysiological characterization more amenable to patch-clamp studies. The activation of TPCs is almost entirely due to PI35P2 in this study. These two points are potential limitations of the study.

A strength of this study is that they examine the effects of potential inhibitors on both sides of the channels (equivalent to cytoplasmic or lysosomal luminal faces). Finally they use molecular docking analysis to propose sites of drug interactions which coupled with substitution mutations to reveal the importance of particular amino acids around the pore gate region such as H699, explaining why the PIKfyve inhibitors show selectivity for TPC2.

Major points

1. Only Fig1A examines NAADP (the endogenous messenger activating TPCs) regulation of TPCs by calcium imaging, subsequent electrophysiological experiments use the drug TPCA1N, probably proposed to mimic the NAADP/NAADP-binding protein complex. The original calcium traces should be shown in addition to the bar chart. Does NAADP activate TPC1PM or TPC2PM? This may require NAADP binding proteins a such as Lsm12 (a seminal discovery of the authors).

2. In Fig 1F the putative NAADP/BP mimetic TPCA1N is used by not discussed in text/figure legend. Is this in addition to addition of PI35P2 as an activator?

Minor point

1. A final cartoon of the TPC and interaction with lipid and drugs might be helpful.

Reviewer #3 (Remarks to the Author):

General remarks:

1. Why are only PM patch clamp experiments performed? The channels are completely out of context in the PM. Please repeat key experiments using endolysosomal patch clamp. You demonstrated in your Nature Commun, 2021 paper that you are able to do this!
2. The statement that there are no potent TPC blockers available is false. Please check Müller et al., Cell Chem Biol., 2021: SG-094 is a potent blocker of TPCs
<https://pubmed.ncbi.nlm.nih.gov/33626324/>
Also Netcharoensirisuk recently identified additional potent blockers of TPCs:
<https://pubmed.ncbi.nlm.nih.gov/33875769/>
Authors ought to cite these papers and tone down their statements relating to this throughout the MS.
3. Tetrandrine is a very lipophilic molecule, it should not matter on which site it is added. Therefore the results in Fig.1E and F come as a big surprise as they cannot be explained by the chemistry of tet., again endolysosomal patch clamp is strongly recommended here.
4. Fig.1A is lacking relevant information, it is unclear how NAADP is applied, original traces must be shown, similar exp. as in A should be provided for PIP2 and the other way around in Fig.1B, patch clamp data for NAADP should be shown.
5. TPC1 L11A/I12A is not a plasma membrane variant according to published literature, please see <https://pubmed.ncbi.nlm.nih.gov/20880839/> Suppl.
Also, our own results do NOT support this, please see attached document.
6. In several exp. n numbers are very low, sometimes only n = 1! Statistics missing throughout the MS, significance values missing e.g. in Fig.3B

Detailed comments:

Page 3

1. Currently, almost no potent and/or selective antagonists of TPCs have been identified.
Not correct, see: Netcharoensirisuk et al 2021; Müller et al 2021

2. It is unknown if Ned-19 (trans-Ned 19), an NAADP signaling antagonist can directly target TPCs.

However, a direct action of tetrandrine on TPCs remains to be proven

In Sakurai et al both NED19 as well as tetrandrine applied directly can block TPC2 (endolysosomal patch-clamp). Please clarify what you mean with "direct action not proven"? Please perform endolysosomal patch clamp exp. With NED19 and tet.

3. Signaling - typo

Page 4

4. Therefore, we tested whether suppression of PI(3,5)P2 production by application of a Ca²⁺ PIKfyve inhibitor, YM201636, affects NAADP-evoked release. We observed that direct microinjection Ca²⁺ of YM201636 greatly reduced NAADP-evoked elevation in HEK293 cells (Fig. 1A).

I suggest to inject PI(3,5)P2 as a control exp.

5. We observed that YM201636 application from the cytosolic side at sub-micromolar concentrations directly inhibited PI(3,5)P2-induced human TPC2 Na⁺ channel currents (Fig. 1B).

Why not use NAADP together with LSM12? and then look for YM blockage. That could keep consistency in the performed experiments. Zhang et al 2021 doi:10.1038/s41467-021-24735-z

6. YM201636's inhibition of the human TPC2 channel was voltage-independent, as shown by the similar levels of inhibition at both negative and positive voltages (Fig. 1D).

Used solutions are not indicated in MM section.

7. Paragraph "Ca²⁺ PIKfyve inhibitor YM201636 suppressed NAADP-evoked release via direct inhibition of TPC2 channels" as well as first part of paragraph "YM201636 is a TPC2 open-channel pore blocker" unclear, please rewrite

Page 5

8. TPC2-A-N1 – typo

9. In this experiment, we activated the TPC currents by perfusion of the activator TPC2-A-N1 on the extracellular side.

I suggest TPC2- A1P in this configuration as control exp.

10. When applied from the cytosolic side, PI(3,5)P2 activated human TPC2 channels with an observed activation rate tau_{on}, of ~10 s, and the effect could be washed off within 1-2 minutes with a deactivation rate of a tau_{off} of ~20 s (Fig. 2A).

Please define and explain tau better.

11. We found that the time courses of TPC2 currents activated by PI(3,5)P2 were similar in the absence and presence of the pre-application of YM21636 (Fig. 2A, C, and D),

Please provide exact values in text and statistical analysis here (which stats test was applied, significance values?)

12. excised patches

Used configuration is not clearly stated, confusing for reader not familiar with patch-clamp

13. We found that the residual effect of YM201636 on PI(3,5)P2-induced channel activation after perfusion with the bath solution alone was washing time-dependent. The residual inhibitory effects

were estimated to be 58%, 33%, and 19% after a 40-s, 80-s and 120-s wash, respectively (Fig. 2H), a time course that was similar to that observed for the inhibition left when YM201636 was washed off in the presence of PI(3,5)P2 (Fig. 2B).

I would like to see a control exp. with other antagonists, e.g. SG-094.
Statistical analysis missing.

Page 6

14. However, the double mutant channel L690A/L694A produced Na⁺ currents in the absence of any agonist, and application of PI(3,5)P2 did not increase the currents (Fig. 2I), indicating that the channels are already constitutively fully open. This result provides functional evidence that these two residues are indeed involved in the formation of the activation gate.

I suggest to examine TPC2- A1N effect too since you are including NAADP block by YM.

15. Compared to the L690A/L694A double mutation, the Ala-substitution of residues inside the channel pore by mutations N305A, T308A, S682A, V686A, and N687A showed little or less effect on TPC2 inhibition by 1 μM YM201636 (Fig. 3B).

Please provide a statistical analysis.

16. The Y312A mutation significantly increased the IC₅₀ for TPC2 inhibition by YM2.1636, by more than 4-fold (IC₅₀ = 0.67 μM) (Fig. 3C and F). The H699A mutation drastically reduced the channel's sensitivity to YM201636, as indicated by a more than 20- fold increase in the IC₅₀ (IC₅₀ = 4.33 μM) compared to that of the wild-type channel (Fig. 3D and F). The double mutation Y312A/H699A resulted in a much greater loss of the channel's sensitivity to YM201636, with an IC₅₀ beyond the highest tested concentration (21 μM) (Fig. 3E and F);

Not clear, please rephrase.

Page 7

17. These results indicate that the cytosolic-side pore-gate formation of L690, L694,

Fig.3b does not really support that, no statistical analysis

18. Paragraph "His699 residue near-pore entrance underlies much greater sensitivity to YM201636 in TPC2 than TPC1"

What about other species?

Page 13

19. Poorly described methodology for performed patch clamp experiments. Please provide more details.

Page 16

Fig 1F - Data acquired in whole cell configuration should be normalized to cell size.

Fig 1 D, E – Please provide higher number of experiments ($n > 3$)

Page 17

Fig 2 A-C, E-G, - At which voltage data were acquired?

Many factors can impair this kind of analysis in A-H. Buffer flow, cell location ...

Are data sets (red to orange fig2 D) acquired from one cell ?

HEK293 cells, 63x

- deletion of combined putative LTS motifs were not sufficient to localize hTPC1 to the plasma membrane (in line with Brailoiu et al. 2010)

Responses to Reviewers' Comments

Reviewer #1 (Remarks to the Author):

The manuscript reports TPC blockers and their action mechanisms. The authors accidentally found that YM201636 and apilimod are different in inducing HEK293 cells' response to NAADP. Then using inside-out patch-clamp recording, they revealed that YM201636 directly acted on TPC2. The authors further suggested that YM201636 is a TPC2 open-channel pore blocker and revealed residues critical for TM201636 binding. H699 is a determinant for the greater sensitivity to YM201636 in TPC2 than TPC1. The study also showed that PI-103, a YM201636 analog, also is a TPC2 blocker. Lastly, computations examined the action mechanism of YM201636 on the channel pore. Although I think that the finding of PI-103 does not contribute significantly to the study and it is better to report an analog active to TPC2 whereas inactive to phosphoinositide kinase, I still recognize the manuscript as a systematic and nice work. And the manuscript has been well written. I recommend to publish the paper as it is.

We highly appreciate the reviewer's support for publication of this manuscript. We agree that an analogue of YM201636 that is specific to TPC2 (not other proteins) will be more desirable than PI-103 which also targets some other enzymes. Our inclusion of PI-103 does provide helpful information about the relevance of the chemical structures to the efficacy of inhibition, and additionally an ion channel target for this research drug that is also commonly used in research in targeting other proteins. The current work provides a new molecular template for future screening or design of a TPC2-specific blocker.

Reviewer #2 (Remarks to the Author):

This is an excellent and convincing contribution from the lab of Jiusheng Yan on the pharmacology of mammalian two pore-channels (TPCs). The study is largely an electrophysiological one investigating the effects of drugs proposed to block these channels and the important discovery that PIKfyve inhibitors are potent open-state inhibitors of TPC2. This latter point is particularly important since these inhibitors are widely employed to reduce the synthesis of PI35P2, an endolysosomal inositol lipid, are implicated as potential anti-cancer and anti-viral drugs, and a modulator itself of endolysosomal cation channels including TPCs.

The approach used is to use mutated channels that target to the plasma membrane rather than their natural loci in the endolysosomal system, thus making their electrophysiological characterization more amenable to patch-clamp studies. The activation of TPCs is almost entirely due to PI35P2 in this study. These two points are potential limitations of the study.

A strength of this study is that they examine the effects of potential inhibitors on both sides of the channels (equivalent to cytoplasmic or lysosomal luminal faces). Finally they use molecular docking analysis to propose sites of drug interactions which coupled with substitution mutations to reveal the importance of particular amino acids around the pore gate region such as H699, explaining why the PIKfyve inhibitors show selectivity for TPC2.

We greatly thank the reviewer for appreciation of the value of this study. We added electrophysiological recording the TPC2 activated by NAADP in whole cell recording and by PI(3,5)P2 in whole lysosome recording. The new results showed that YM201636 is effective in inhibition of TPC2 activity in a lysosomal environment and NAADP-evoked TPC2 activation. We demonstrated that YM201636, as expected for a pore blocker, inhibits TPC2 currents in a manner independent of ligands and environment.

Major points

1. Only Fig1A examines NAADP (the endogenous messenger activating TPCs) regulation of TPCs by calcium imaging, subsequent electrophysiological experiments use the drug TPCA1N, probably proposed to mimic the NAADP/NAADP binding protein complex. The original calcium traces should be shown in addition to the bar chart. Does NAADP activate TPC1PM or TPC2PM? This may require NAADP binding proteins a such as Lsm12 (a seminal discovery of the authors).

As suggested, we added representative calcium traces. We added data to show that YM201636 also blocks NAADP-evoked TPC2 currents.

2. In Fig 1F the putative NAADP/BP mimetic TPCA1N is used by not discussed in text/figure legend. Is this in addition to addition of PI35P2 as an activator?

We revised here to describe TPC2-A1-N and its application. TPC2-A1-N was used alone without PI(3,5)P2.

Minor point

1. A final cartoon of the TPC and interaction with lipid and drugs might be helpful.

We respectfully request to not provide such a cartoon because of two reasons. The TPC2-blockade by YM201636 is independent of the ligand, include the PI(3,5)P2. Thus, to be concise and for clarity, all cartoons in figures display the structures of the pore region only. From molecular docking and molecular dynamic simulation, the binding of YM201636 in the pore region appeared to be dynamic and currently hard to choose a specific binding mode to highlight.

Reviewer #3 (Remarks to the Author):

We greatly appreciate the reviewer's thorough review and constructive comments. As advised, we added electrophysiological data of TPC2 activated by NAADP in whole cell recording and by PI(3,5)P2 in whole lysosome recording. The new results consolidate that YM201636, as expected for a pore blocker, inhibits TPC2 currents in a manner independent of ligands and environment.

General remarks:

1. Why are only PM patch clamp experiments performed? The channels are completely out of context in the PM. Please repeat key experiments using endolysosomal patch clamp. You demonstrated in your Nature Commun, 2021 paper that you are able to do this!

We preferred to do the PM patch clamp experiment because it was much convenient to do and the high data quality of inside-out patch due to the small size of membrane clamped. We now added lysosomal recording to confirm that the YM201636 acts as an effective TPC2 blocker regardless of membrane environment. The PM TPC2 mutant has been used by many labs. We agree that the channels need to be naturally expressed on the endolysosomal system if the purposes are to study the channel's cellular and physiological function or endolysosomal-specific channel properties or regulation. However, we saw advantage and no obvious drawback in using PM TPC2 for general pharmacological and biophysical studies.

2. The statement that there are no potent TPC blockers available is false. Please check Müller et al., Cell Chem Biol., 2021: SG-094 is a potent blocker of TPCs

<https://pubmed.ncbi.nlm.nih.gov/33626324/>

Also Netcharoensirisuk recently identified additional potent blockers of TPCs:

<https://pubmed.ncbi.nlm.nih.gov/33875769/>

Authors ought to cite these papers and tone done their statements relating to this throughout the MS.

Thanks for pointing out the newly identified TPC inhibitors! We have included them in the introduction and made changes in introduction and discussion.

3. Tetrandrine is a very lipophilic molecule, it should not matter on which site it is added. Therefore the results in Fig.1E and F come as a big surprise as they cannot be explained by the chemistry of tet., again endolysosomal patch clamp is strongly recommended here.

We agree that a lipophilic modulator can act similarly from both sides in whole cell recording condition if given enough time for the molecule to equilibrate across membrane. However, under perfusion condition for drug

application within a limited time, the molecule might not be able to reach same concentration inside cells as the outside solution. For patch-clamp recording of excised membrane patch, it particularly matters which side the drug is applied as one side has constant concentration while other side will have dilution effect once the molecule come from the other high (fixed) concentration side, unless the site is equally accessible in deep membrane from both sides.

As we mentioned below, we decided to remove the part related to tetrandrine.

4. Fig.1A is lacking relevant information, it is unclear how NAADP is applied, original traces must be shown, similar exp. as in A should be provided for PIP2 and the other way around in Fig.1B, patch clamp data for NAADP should be shown.

The details are described in method. We added “microinjection” in figure legend. We added representative traces (new Fig 1). We added NAADP-induced TPC2 currents (Fig 1C and D).

5. TPC1 L11A/I12A is not a plasma membrane variant according to published literature, please see <https://pubmed.ncbi.nlm.nih.gov/20880839/> Suppl. Also, our own results do NOT support this, please see attached document.

Previously, the TPC1 L11A/I12A plasma membrane currents had been well documented (<https://pubmed.ncbi.nlm.nih.gov/29562233/>). We observed that the TPC1-L11A/I12A plasma membrane currents were generally not as large as those from TPC2-L11A/L12A. But, we had no major issue in recording the currents. Patch-clamp recording is more sensitive than the method of protein localization imaging in detecting the ion channels.

6. In several exp. n numbers are very low, sometimes only n = 1! Statistics missing throughout the MS, significance values missing e.g. in Fig.3B

We added statistic information in all figures and/or legends. For high data quality large difference and qualitative purpose, we consider n of 3 to be reasonable for an evaluation of the data reproducibility unless the variation is big. Current traces shown are representative. Except the molecular docking, all experiments have repeats (n≥3).

Detailed comments:

Page 3

1. Currently, almost no potent and/or selective antagonists of TPCs have been identified.

Not correct, see: Netcharoensirisuk et al 2021; Müller et al 2021

We have included them in the introduction and made changes in introduction and discussion.

2. It is unknown if Ned-19 (trans-Ned 19), an NAADP signaling antagonist can directly target TPCs.

However, a direct action of tetrandrine on TPCs remains to be proven

In Sakurai et al both NED19 as well as tetrandrine applied directly can block TPC2 (endolysosomal patch-clamp). Please clarify what you mean with “direct action not proven”? Please perform endolysosomal patch clamp exp. With NED19 and tet.

We think the topics of Ned-19 and tetrandrine are important and deserve more detailed investigation to clarify given their popularity. Because they are very marginal to this manuscript which is focused on YM201636 and its analogue, we removed parts related to Ned-19 and tetrandrine in results and discussion. This deletion has no impact on quality and conclusion of the manuscript.

3. Signaling - typo

Corrected

Page 4

4. Therefore, we tested whether suppression of PI(3,5)P2 production by application of a Ca²⁺ PIKfyve inhibitor,

YM201636, affects NAADP-evoked release. We observed that direct microinjection Ca^{2+} of YM201636 greatly reduced NAADP-evoked elevation in HEK293 cells (Fig. 1A). I suggest to inject PI(3,5)P2 as a control exp.

PI(3,5)P2 had been used in most electrophysiological experiments in this work. We don't feel that injection of PI(3,5)P2 provides additional information about YM201636 blockade of TPC2. We are not sure PI(3,5)P2 can be a good negative control for Ca^{2+} release as it can have other lysosomal protein targets. Therefore, we respectfully don't agree to add this experiment.

5. We observed that YM201636 application from the cytosolic side at sub-micromolar concentrations directly inhibited PI(3,5)P2-induced human TPC2 Na^+ channel currents (Fig. 1B).

Why not use NAADP together with LSM12? and then look for YM blockage. That could keep consistency in the performed experiments. Zhang et al 2021 doi:10.1038/s41467-021-24735-z

The proposed experiment is worthy doing but beyond the focus of this work. We have demonstrated that its action is independent of ligands and environment. Therefore, we respectfully don't agree to include the proposed experiment in this manuscript.

6. YM201636's inhibition of the human TPC2 channel was voltage-independent, as shown by the similar levels of inhibition at both negative and positive voltages (Fig. 1D).

Used solutions are not indicated in MM section.

We added detail in method and legend. We added description for the recording configuration and used solutions for all figures in figure legends.

7. Paragraph " Ca^{2+} PIKfyve inhibitor YM201636 suppressed NAADP-evoked release via direct inhibition of TPC2 channels" as well as first part of paragraph "YM201636 is a TPC2 open-channel pore blocker" unclear, please rewrite Page 5

We have revised these 2 parts to improve readiness and understanding.

8. TPC2-A-N1 – typo

Corrected

9. In this experiment, we activated the TPC currents by perfusion of the activator TPC2-A-N1 on the extracellular side. I suggest TPC2- A1P in this configuration as control exp.

Given that we have demonstrated that YM201636's blockade of TPC2 is independent of ligands and environment, we feel it is not necessary to add more activators. Therefore, we respectfully don't agree to add the proposed experiment.

10. When applied from the cytosolic side, PI(3,5)P2 activated human TPC2 channels with an observed activation rate tauon, of ~ 10 s, and the effect could be washed off within 1-2 minutes with a deactivation rate of a tauoff of ~ 20 s (Fig. 2A). Please define and explain tau better.

We added definition of Tau after it was first mentioned in the text.

11. We found that the time courses of TPC2 currents activated by PI(3,5)P2 were similar in the absence and presence of the pre-application of YM201636 (Fig. 2A, C, and D), Please provide exact values in text and statistical analysis here (which stats test was applied, significance values?)

Changed as suggested.

12. excised patches

Used configuration is not clearly stated, confusing for reader not familiar with patch-clamp

We added recording configuration for all data in figure legends.

13. We found that the residual effect of YM201636 on PI(3,5)P2-induced channel activation after perfusion with the bath solution alone was washing time-dependent. The residual inhibitory effects were estimated to be 58%, 33%, and 19% after a 40-s, 80-s and 120-s wash, respectively (Fig. 2H), a time course that was similar to that observed for the inhibition left when YM201636 was washed off in the presence of PI(3,5)P2 (Fig. 2B).

I would like to see a control exp. with other antagonists, e.g. SG-094.

Comparison of YM201636 with other inhibitors is helpful but not the focus of this work. Therefore, we respectfully don't agree to add other inhibitors to test.

Statistical analysis missing.

We added statistical information.

Page 6

14. However, the double mutant channel L690A/L694A produced Na⁺ currents in the absence of any agonist, and application of PI(3,5)P2 did not increase the currents (Fig. 2I), indicating that the channels are already constitutively fully open. This result provides functional evidence that these two residues are indeed involved in the formation of the activation gate.

I suggest to examine TPC2- A1N effect too since you are including NAADP block by YM.

We had used TPC2-A1-N in the whole cell recording to show the effect of YM. If the purpose is to see whether TPC2-A1-N can increase the currents of the constitutively open L690A/L694A mutant channel, we consider it to be beyond the focus of this manuscript. Therefore, we respectfully don't agree to add the proposed experiment.

15. Compared to the L690A/L694A double mutation, the Ala-substitution of residues inside the channel pore by mutations N305A, T308A, S682A, V686A, and N687A showed little or less effect on TPC2 inhibition by 1 μ M YM201636 (Fig. 3B).

Please provide a statistical analysis.

We added statistical information.

16. The Y312A mutation significantly increased the IC₅₀ for TPC2 inhibition by YM2.1636, by more than 4-fold (IC₅₀ = 0.67 μ M) (Fig. 3C and F). The H699A mutation drastically reduced the channel's sensitivity to YM201636, as indicated by a more than 20- fold increase in the IC₅₀ (IC₅₀ = 4.33 μ M) compared to that of the wild-type channel (Fig. 3D and F). The double mutation Y312A/H699A resulted in a much greater loss of the channel's sensitivity to YM201636, with an IC₅₀ beyond the highest tested concentration (21 μ M) (Fig. 3E and F);

Not clear, please rephrase.

Page 7

We rephrased and hope it is better now.

17. These results indicate that the cytosolic-side pore-gate formation of L690, L694, Fig.3b does not really support that, no statistical analysis

We added statistic information (new Fig 4E). The averaged decrease in inhibition by 1 uM YM201636 caused by L690A/L694A is higher than other mutations deeper in the pore (new Fig 4E). The IC50 is increased more than 3 folds by this double mutation (Fig 4D), which we considered to be significant.

18. Paragraph “His699 residue near–pore entrance underlies much greater sensitivity to YM201636 in TPC2 than TPC1”
What about other species?

We added a sentence in discussion (end of the first paragraph) to indicate that His699 is conserved in TPC2 among species.

Page 13

19. Poorly described methodology for performed patch clamp experiments. Please provide more details.

We added more details for patch-clamp experiments in methods.

Page 16

Fig 1F - Data acquired in whole cell configuration should be normalized to cell size.

Since the data was from the same cell (new Fig 2G), it should be ok to use the current directly.

Fig 1 D, E – Please provide higher number of experiments ($n > 3$)

We consider that $n=3$ reasonably good given the high quality of the data (large currents and small variation) in many figures of this work. If we are looking for small difference and the difference is important, a high n number is definitely needed.

Page 17

Fig 2 A-C, E-G, - At which voltage data were acquired?

We added the voltage used.

Many factors can impair this kind of analysis in A-H. Buffer flow, cell location ...

Are data sets (red to orange fig2 D) acquired from one cell ?

We understand the concern and we are confident in data. Each data set was acquired from the same cell.

Additional changes made:

As new data and figures have been added, we reorganized figures and text (Fig 1- 4) to improve the logics and readiness.

Reviewers' comments:

Reviewer #2 (Remarks to the Author):

I am generally happy with the responses that the authors have provided to the first round of reviews.

However, I have one fundamental query. This concerns the experiments with NAADP that were missing from the original study (Figs 1C,D).

Why is NAADP microinjected when measuring TPC currents and not simply perfused into the cell via the patch pipet. I found this confusing.

Could the authors please elaborate.

Reviewer #3 (Remarks to the Author):

1) Why do the currents in Fig1C show inward rectification?

2) Line 60: not sure if citations are correct here? (should be 30 instead of 36?); also citations 21 and 37 are duplicates!

60% at 0.5 and 54% at 10 uM is not necessarily a discrepancy. If 0.5 is already saturating one would not expect that 10 uM has a much bigger effect. 60% versus 54% could be considered the same within error ranges.

I recommend to rather argue here that tetrandrine is not TPC specific and also blocks many other targets, hence cannot be considered an optimal TPC blocker

Minor points:

Please rewrite text in line 36-38:

TPCs are homodimeric cation channels. Each subunit contains ... TPCs are potently activated

Line 50:

Suggested to rewrite as follows:

Potent and/or selective modulators are important pharmacological tools in understanding molecular mechanisms and physiological and pathological function of an ion channel.

Line 52-53:

Suggested to rewrite as follows:

Ned-19 (trans-Ned 19) is commonly used as an NAADP signaling antagonist albeit also blocking PI(3,5)P2 currents (30) and shown to form a complex with plant TPC1 (solved X-ray structure)

(Comment: Ned19 does demonstrably block not only NAADP but also PI(3,5)P2, see Sakurai et al. Suppl.)

Line 55:

inhibition (~75%) of lysosomal TPC2

Line 62:

Write: flavonoid NOT favonoid

Line 114ff:

Suggested to rewrite as follows:

The observed reduced inhibition under this condition as compared to when it was applied on the cytosolic side ...equilibration of the chemical across the membrane during perfusion and also dilution by the pipette solution once the inhibitor is inside the cell.

Responses to Reviewers' Comments

Reviewer #2 (Remarks to the Author):

I am generally happy with the responses that the authors have provided to the first round of reviews. However, I have one fundamental query. This concerns the experiments with NAADP that were missing from the original study (Figs 1C,D).

Why is NAADP microinjected when measuring TPC currents and not simply perfused into the cell via the patch pipet. I found this confusing.

Could the authors please elaborate.

This is an interesting question. We had previously developed and reported this NAADP microinjection method for measuring NAADP-activated TPC currents (Ref 18). In that report, we had speculated why this method worked and pipette diffusion failed. The main difference is that microinjection is a quick delivery while the diffusion is a slow process. One possibility is that the NAADP-induced current is transient, e.g., caused by channel inactivation. Then, the slow diffusion process cannot generate large transient currents for recording. Since this had been discussed in our previous report, we feel it is not necessary to re-discuss it in this manuscript.

Reviewer #3 (Remarks to the Author):

1) Why do the currents in Fig1C show inward rectification?

We prefer not to discuss about the current shape. We are confident that the majority of the currents were TPC currents as we included multiple negative controls in the previous reports and here YM201636 can block the currents. For recorded whole cell currents that are relatively small (below nA), the current shapes (even after subtraction from baseline currents) can be affected by small drift in leak currents and system imperfection in resistance and capacitance compensation. Therefore, we hesitate to discuss the shape of the currents as we don't want to overinterpret.

2) Line 60: not sure if citations are correct here? (should be 30 instead of 36?); also citations 21 and 37 are duplicates!

Corrected.

60% at 0.5 and 54% at 10 μM is not necessarily a discrepancy. If 0.5 is already saturating one would not expect that 10 μM has a much bigger effect. 60% versus 54% could be considered the same within error ranges.

I recommend to rather argue here that tetrandrine is not TPC specific and also blocks many other targets, hence cannot be considered an optimal TPC blocker

Revised as suggested (removed the statement about discrepancy and rewrite as suggested).

Changed to "Tetrandrine, a voltage-gated Ca^{2+} (Ca_V) channel blocker, was reported to inhibit Ebola virus entry into host cells presumably via inhibition of TPCs³⁰. However, tetrandrine is hardly an optimal TPC antagonist because of its issue in specificity and currently the lack of reported full inhibition of lysosomal TPC2 currents, e.g., 50-60% inhibition by 0.5 μM ³⁰ and 10 μM tetrandrine²¹, in spite of its potent effect on virus entry (IC_{50} of 55 nM)³⁰."

Minor points:

Please rewrite text in line 36-38:

TPCs are homodimeric cation channels. Each subunit contains ... TPCs are potently activated

Revised as suggested.

Changed to “TPCs are homodimeric cation channels. Each subunit contains two transmembrane domains of the basic structural unit (six transmembrane segments and a pore loop) of a voltage-gated ion channel. TPCs are potently activated by phosphatidylinositol 3,5-bisphosphate (PI(3,5)P₂)³⁻⁶, inhibited by ATP via mTORC1⁷, and slightly blocked by cytoplasmic and luminal Mg²⁺⁵.”

Line 50:

Suggested to rewrite as follows:

Potent and/or selective modulators are important pharmacological tools in understanding molecular mechanisms and physiological and pathological function of an ion channel.

Revised as suggested.

Changed to “Potent and/or selective modulators are important pharmacological tools in understanding molecular mechanisms and physiological and pathological function of an ion channel.”

Line 52-53:

Suggested to rewrite as follows:

Ned-19 (trans-Ned 19) is commonly used as an NAADP signaling antagonist albeit also blocking PI(3,5)P₂ currents (30) and shown to form a complex with plant TPC1 (solved X-ray structure)

(Comment: Ned19 does demonstrably block not only NAADP but also PI(3,5)P₂, see Sakurai et al. Suppl.)

Revised similarly as suggested.

Changed to “Ned-19 (trans-Ned 19) is a commonly used as an NAADP signaling antagonist³³, albeit it also blocked PI(3,5)P₂-induced TPC current³⁰ and formed complex with plant TPC1 in solved X-ray structure³⁴. The potency of Ned-19 in TPC inhibition remains unclear as only a high concentration (200 μM) of Ned-19 was reported to be associated with a significant inhibition (~75%) lysosomal TPC2 activity³⁰.”

Line 55:

inhibition (~75%) of lysosomal TPC2

Changed as suggested to “inhibition (~75%) lysosomal TPC2 activity”.

Line 62:

Write: flavonoid NOT favonoid

Corrected.

Line 114ff:

Suggested to rewrite as follows:

The observed reduced inhibition under this condition as compared to when it was applied on the cytosolic side ...equilibration of the chemical across the membrane during perfusion and also dilution by the pipette solution once the inhibitor is inside the cell.

Changed as suggested to “The observed reduced inhibition under this condition as compared to when it was applied on the cytosolic side in inside-out configuration likely suggests favorable accessibility of the inhibitory site from the cytosolic side. The concentration of the inhibitor could be lower inside cells than the outside solution, caused by insufficient equilibration of the chemical across membrane during perfusion and also dilution by the pipette solution once the inhibitor is inside the cell.”